# Multistage Nanocarrier Based on an Oil Core–Graphene Oxide Shell

**DOI:** 10.3390/pharmaceutics16060827

**Published:** 2024-06-18

**Authors:** Immacolata Tufano, Raffaele Vecchione, Valeria Panzetta, Edmondo Battista, Costantino Casale, Giorgia Imparato, Paolo Antonio Netti

**Affiliations:** 1Center for Advanced Biomaterials for Health Care (CABHC), Istituto Italiano di Tecnologia, 80125 Naples, Italy; 2Department of Chemical Materials and Industrial Production (DICMAPI), University of Naples Federico II, 80138 Naples, Italy; 3Interdisciplinary Research Centre on Biomaterials (CRIB), University of Naples Federico II, 80138 Naples, Italy; 4Department of Innovative Technologies in Medicine & Dentistry, University “G. d’Annunzio” Chieti-Pescara, Via dei Vestini, 66100 Chieti, Italy

**Keywords:** graphene oxide quantum dots, oil/water nano-emulsion, multi-stage approach

## Abstract

Potent synthetic drugs, as well as biomolecules extracted from plants, have been investigated for their selectivity toward cancer cells. The main limitation in cancer treatment is the ability to bring such molecules within each single cancer cell, which requires accumulation in the peritumoral region followed by homogeneous spreading within the entire tissue. In the last decades, nanotechnology has emerged as a powerful tool due to its ability to protect the drug during blood circulation and allow enhanced accumulation around the leaky regions of the tumor vasculature. However, the ideal size for accumulation of around 100 nm is too large for effective penetration into the dense collagen matrix. Therefore, we propose a multistage system based on graphene oxide nanosheet-based quantum dots (GOQDs) with dimensions that are 12 nm, functionalized with hyaluronic acid (GOQDs-HA), and deposited using the layer-by-layer technique onto an oil-in-water nanoemulsion (O/W NE) template that is around 100 nm in size, previously stabilized by a biodegradable polymer, chitosan. The choice of a biodegradable core for the nanocarrier is to degrade once inside the tumor, thus promoting the release of smaller compounds, GOQDs-HA, carrying the adsorbed anticancer compound, which in this work is represented by curcumin as a model bioactive anticancer molecule. Additionally, modification with HA aims to promote active targeting of stromal and cancer cells. Cell uptake experiments and preliminary penetration experiments in three-dimensional microtissues were performed to assess the proposed multistage nanocarrier.

## 1. Introduction

The National Cancer Institute identified nanotechnology as the science that has the potential to significantly modify the basis for the diagnosis, treatment, and prevention of cancer. Indeed, nanotechnology allows for cancer investigation and treatment at the molecular level, in real-time, and in the early stages of the process [1]. The classic application of nanotechnology in cancer therapy involves enhancing pharmacokinetics and reducing the side effects of chemotherapy by using nanocarriers to deliver anticancer therapeutics to tumor tissues with selective targeting [2]. However, an ideal nanocarrier should be able to provide the anticancer drug to the entire tumor tissue. Nanoparticles of around 100–200 nanometers in size can accumulate preferentially in solid tumors by exploiting the altered vascularization and reduced lymphatic drainage typical of a growing tumor mass (enhanced permeability and retention EPR effect), thus reducing normal tissue toxicity. However, their diffusion in the dense collagen matrix of the interstitial space is weak, resulting in nanoparticle aggregation around the tumor’s blood vessel and limited penetration into the tumor parenchyma [3]. Conversely, 10–30 nm nanoparticles can more easily penetrate tissues and show high cellular uptake but they can be easily sequestered by the mononuclear phagocytic system and eliminated during their path. Therefore, Wong et al. demonstrated that a multistage approach enables diffusion into the dense collagen matrix and penetration within the tumor interstitial space based on systems able to accumulate in tumor tissues via the EPR effect, and then achieve deeper penetration through size shrinking. The multistage approach supports many next-generation drug delivery systems, predominantly composed of metal oxide nanoparticles [4,5]. Among the diverse nanocarriers for cancer drug delivery studied in the last decades, particular interest has been shown in layer-by-layer (LbL) polymeric nanocapsules, mainly based on electrostatic interactions between thin polymer layers [6,7,8]. Since 2014, our team developed biodegradable oil-in-water nanoemulsions (O/W NEs, size down to 100 nm) coated with biodegradable polyelectrolytes via the LbL technique, with high monodispersity and stability over time, even without using additional stabilizing agents [9]. This system offers several pharmaceutical and therapeutic benefits, including ease of synthesis, scalable production, good biocompatibility, biodegradability, reasonable stability, and the ability to incorporate several hydrophobic drugs, as well as lipophilic biomolecules [10,11,12,13] and inorganic contrast agents [14,15]. In this study, to provide a multistage system based on the same multi-size principles described in [3], we combine our 100 nm O/W NEs, via LbL approach, with nano-graphene oxide quantum dots (nGO or GOQDs) meant for deep tissue penetration due to their small size (of up to 10 nm) upon NE degradation. GOQDs were chosen for their unique proprieties, making them one of the most promising nanoparticles for nanomedicine applications [16]. GOQDs possess a large number of reactive functional groups allowing for modification to improve stability, solubility, and biocompatibility [17]. Furthermore, the large surface area containing aromatic crystalline regions and amorphous sp^3^ hybridized domains allows for high loading of hydrophobic molecules via van der Waals or π-π interactions, making GO a good candidate for the delivery of water-insoluble aromatic anticancer drugs [18]. Additionally, when GO is reduced to dimensions of around tens of nm, it also exhibits photoluminescence [19]; this, besides enabling it as an imaging agent, allows its use as an agent for photothermal therapy, a minimally invasive treatment based on the induction of cellular hyperthermia [20]. In this complex system, the GOQDs were functionalized with amino groups through oxidative cutting promoted by hydrogen peroxide and ammonia, and their surface was covalently coated with hyaluronic acid to increase the biocompatibility of the GOQDs and to obtain an active targeting system. Active targeting is usually performed by binding a variety of ligands to the surface of the nanocarrier to recognize specific surface molecules that are overexpressed by tumor cells but not present in normal cells [21]. HA, indeed, is a naturally occurring, biocompatible, non-immunogenic, and biodegradable polysaccharide able to selectively recognize transmembrane glycoprotein CD44 receptors, which are highly expressed on the surface of tumor cells [22,23,24,25]. O/W NEs were coated with a layer of chitosan, a biodegradable polymer, to form stable and surface-positively charged secondary emulsions (Ct-NEs), on which negatively charged GOQDs-HA could be deposited. The complete system exhibited high stability in the monitoring window of 180 days from its preparation and high biocompatibility. Curcumin, a naturally occurring multifunctional polyphenolic phytoconstituent, was selected as a hydrophobic therapeutic model to analyze the loading and release capacity of pharmacologically active molecules from the system. Even though the target of GOQD-HA-coated Ct-NEs is represented by tumor cells, in many solid tumors, such as breast adenocarcinoma, therapeutics show limited efficacy due to their limited penetration into the tumor stroma, composed of an abundant extracellular matrix and various stroma cells, such as fibroblasts [26,27]. Based on this consideration, it is relevant to assess the ability of GOQDsHA-coated Ct-NEs to interact with both tumor and stroma cells and to penetrate the tumor stroma surrounding tumor cells. To this end, a biological assessment was performed by exploiting the photoluminescence of the obtained systems in three types of cell lines, human dermal fibroblasts (HDFs), mammary epithelial (MCF10A), and adenocarcinoma cells (MDA-MB-231). Preliminary results on the multistage approach were obtained by performing tissue penetration experiments of both free GOQDs-HA and GOQD-HA-coated Ct-Nes, where in both cases, GOQD was labeled with Rhodamine B (Rhod) for fluorescence microscopy analysis. It appears that over time the penetration of GOQDs-HA, whether free or loaded on Ct-Nes, was similar, as expected in the cases of degradation and deconstruction of the multistage nanocarrier, followed by the release of the smaller nanoparticles based on GOQDs-HA.

## 2. Methods

### 2.1. Materials

Graphite (flakes, 325 mesh), 30% sulfuric acid (H_2_SO_4_ 95–98% wt), potassium permanganate (KmnO_4_) hydrogen peroxide (H_2_O_2_ 30% *w*/*v*), ammonium solution (28–30% *w/v* NH_3_), N-hydroxysuccinimide (NHS), 1-(3-Dimethylaminopropyl)-3-ethyl carbodiimide hydrochloride (EDC), curcumin (from Curcuma longa turmeric, powder), Rhodamine B isothiocyanate, fluorescein isothiocyanate (FITC), chitosan (LMW 90–50 kDa,) minimum essential medium (MEM), fetal bovine serum (FBS), Eagle’s minimal essential medium (EMEM), and 1% penicillin/streptomycin were purchased from Sigma-Aldrich (Milano, Italy). Hyaluronic acid 50 kDa was purchased from HAWORKS (Bedminster, NJ, USA). 1% of L-glutamine was obtained from Lonza 17-605E, (Basel, Switzerland); 0.25% of trypsin and 1 mM EDTA were purchased from Microtech (Napoli, Italy). Dialysis membranes were purchased from Spectrum Laboratories Inc. (New Brunswick, NJ, USA). Amicon Ultra centrifugal filter units (3 kDa) were purchased from Merck Millipore (Milano, Italy). Polystyrene tissue culture flasks, 150 cm^2^, were obtained from Corning Inc. (Corning, NY, USA). O/W NEs were prepared using soybean oil (density of 0.922 g/mL at 20 °C) and the surfactant Lipoid E80 (an egg lecithin powder enriched with 80–85% phosphatidylcholine (PC) and 7–9.5% phosphatidyl ethanolamine (PE)) purchased from Lipoid (Ludwigshafen, Germany). Millipore^®^ Milli-Q water was used to prepare all NEs and solutions. All the reagents were used as received.

### 2.2. Instrumentation

Cryo-TEM analysis was performed using a Tecnai G2 F20 transmission electron microscope (FEI Company, The Netherlands) equipped with a Schottky field emission gun operating at an acceleration voltage of 200 kV. Images were recorded at low doses with a 2 k × 2 k UltraScan (Gatan, Pleasanton, CA, USA) CCD camera. Frozen hydrated samples were prepared by applying a 3 μL aliquot to a previously glow-discharged 200 mesh holey carbon grid (Ted Pella, Redding, CA, USA). Before plunging into nitrogen-cooled liquid ethane, the grid was blotted for 1.5 s in a chamber at 4 °C and 90% humidity using a FEI Vitrobot Mark IV (FEI Company, The Netherlands). Purity and surface chemical composition were determined by energy-dispersive X-ray spectroscopy (EDX). A 20 μL sample of GOQDs-NH_2_ or GOQDs-HA (1 mg/mL) was deposited onto a standard SEM pin stub and analyzed by FESEM ULTRA-PLUS (Zeiss) (Milan, Italy) with the SE2 detector. Powder X-ray diffraction (XRD) patterns were collected on GO-QDs samples using a Philips PW1710 apparatus with CuKα1 radiation. The scanning step size was 0.010° in 2θ, and the time for each step was 1 s. Field emission scanning electron microscopy (SEM) micrographs were collected with a Zeiss Ultra Plus system. The thickness of the samples was measured on a mica surface using a Multimode NanoScope V scanning probe microscopy system (Bruker, Billerica, MA, USA) with AFM cantilever tips having a force constant of ∼50 N/m and a resonance vibration frequency of ∼350 kHz (Bruker, USA). The samples were prepared using a solution casting of the aqueous suspensions of GO on a freshly cleaved mica surface and drying in air. Chemical information was revealed by X-ray photoelectron spectroscopy (XPS) using a PHI5000VersaProbeII XPS spectrometer with a monochromatic Al-K-α source of 1486.68 eV. High-resolution C1s spectra were acquired at high power (100 W) with a pass energy of 23.5 eV, 0.1 eV step size, and averaged over 20 scans. Spectra from insulating samples were corrected by shifting all peaks to the adventitious carbon C1 spectral component binding energy set at 284.8 eV. Multipack 8.2 software was used to process all the spectra. The ultraviolet-visible (UV/Vis) spectrum of the sample was measured using a Cary 5000 UV/Vis spectrophotometer. Photoluminescence spectra were obtained using an LS55 spectrophotometer (Perkin Elmer, Milano, Italy). Nanoparticle tracking analysis (NTA) measurements were performed in a NanoSight NS300 (Malvern Panalytical Ltd., Malvern, UK), equipped with a 488 nm laser. For the NTA measurements, the bilayer solution was diluted in deionized water (1:4000). Before analysis, the solution was homogenized for 3 min in an ultrasonic bath. Samples were injected into the chamber with sterile syringes. The size distribution and ζ potential in the solution were detected using a dynamic light scattering (DLS) instrument (Zetasizer nano ZS series ZEN 3600, Malvern Instruments Ltd., Malvern, UK, λ 632.8 nm). All the samples were diluted to a droplet concentration of approximately 0.025 wt% using milli-Q water. Ζ-potential analysis was carried out by setting 50 runs for each measurement. Confocal fluorescence microscopy images were captured with a Leica TCS SP5. Images were acquired with a field of view of 77.5 × 77.5 μm for a pixel size of 76 × 76 nm and visualized by LAS-AF 3.2 software (Leica-Microsystems, Mannheim, Germany). A laser scanning confocal microscope TCS SP5 SMD (Leica Microsystems, Germany) equipped with a Chameleon Ultra II 80 MHz pulsed NIR laser (Coherent Inc., Santa Clara, CA, USA) was used to perform co-localization experiments. An HCX IRAPO L 25×/0.95 water immersion objective was used for all the experiments, exciting FITC using a multi-photon laser at 740 nm and detecting emission at 500–530 nm, whereas Rhod was excited using a multi-photon laser at 840 nm, and its emission was detected at 560–610 nm. LAS-AF software was used to export data.

### 2.3. Graphene Oxide Preparation

GO was prepared by oxidizing natural graphite powder (325 mesh) according to the Hummers method, with a modification involving the removal of NaNO_3_ from the reaction [28]. Graphite powder (3.0 g) was added to concentrated H_2_SO_4_ (70 mL) while stirring in an ice bath. Under vigorous agitation, KMnO_4_ (9.0 g) was slowly added keeping the temperature of the suspension below 20 °C. The reaction mixture was then transferred to an oil bath at 40 °C and vigorously stirred for about 0.5 h. Then, 150 mL water was added, and the solution was stirred for 15 min at 95 °C. An additional 500 mL water was then added, followed by the slow addition of 15 mL H_2_O_2_ (30%), which changed the solution color from dark brown to yellow. The mixture was filtered and washed with 250 mL of 1:10 HCl aqueous solution to remove metal ions. The resulting solid was dried in air and diluted to 600 mL providing a graphite oxide aqueous dispersion. This dispersion was then purified by dialysis for one week using a dialysis membrane with a molecular weight cut-off of 8000–14,000 Da to remove any remaining metal species. The resultant graphite oxide aqueous dispersion was diluted to 1.2 L, stirred overnight, and sonicated (ultrasonic bath Falc Instruments) at a power of 20 W for 30 min to exfoliate it. The GO dispersion was then centrifuged at 8000 rpm for 40 min to remove unexfoliated graphite and larger GO sheets. The supernatant was collected and lyophilized for two days to yield graphene oxide (GO_Hummer_).

### 2.4. Amino-Modified GOQDs Preparation

Amino-modified GOQDs (GOQDs-NH_2_) were obtained through the oxidative cutting procedure promoted by hydrogen peroxide and ammonia, as reported in the literature, with slight modifications [29]. Briefly, in a 250 mL bottom flask, H_2_O_2_ (40 mL, 30%) was added to GO_Hummer_ dispersion (3 mL, 8 mg/mL in H_2_O), and the temperature was brought to 80° C while stirring in an oil bath. Afterward, ammonia (7 mL, 25–28%) was added dropwise to the mixture over five hours, resulting in vigorous bubbling and heating. After stirring for 8 h, under reflux, a clear, homogenous yellowish-brown mixture was obtained with no detectable or precipitated GO remaining. The prepared mixture was then filtered through a 0.22 μm cellulose acetate membrane, and the volume was reduced by evaporation under vacuum. The resulting solution was dialyzed for two days against deionized water (MWCO 100–500 Da), freeze-dried, and stored at 4 °C in the dark.

### 2.5. Conjugation of GOQDs-NH_2_ with HA

An aqueous HA (1 *w*/*v* %) solution was prepared by dissolving the powder form of HA (0.10 g, 0.24 meq of COOH) in 10 mL of DI water. The pH of the HA solution was adjusted to 4.5. Then, EDC (0.45 g, 2.4 mmol) and NHS (0.14 g, 1.2 mmol) were added to the HA solution at 22 °C. After stirring for 1 h, the GOQDs-NH_2_ solution (0.10 g, 5 mL) was added to the HA solution, and the reaction mixture was continuously stirred overnight. The pH was adjusted to 8−9 to terminate the reaction. The unreacted species were removed from the reaction mixture through dialysis (MWCO 12,000−14,000) against DI water (three times), and the GOQDs-HA conjugated was finally recovered by lyophilization as a brownish solid.

### 2.6. GOQDs-Rhodamine B Labeling

Rhodamine B was used to label GOQDs by physical adsorption; 158.8 µL of Rhod solution, (1 mg/mL in H_2_O) was added to 3.58 mL of GOQDs (0.85 mg/mL in H_2_O). After stirring for 24 h at room temperature in the dark, the Rhod excess was washed by dialysis against water for 2 days in the dark. The amount of physio-absorbed Rhod on GOQDs was determined from a UV-Vis calibration curve of Rhod at 557 nm.

### 2.7. Oil/Water Nanoemulsion (O/W NE) Preparation

The O/W NE was prepared according to a two-step procedure previously developed [9]. The narrowly distributed O/W NE obtained by high-pressure microfluidization is coated with a positively charged chitosan layer, which significantly extends the stability of the system over time. This chitosan-coated NE is referred to as secondary NE. Briefly, the oil phase was prepared by adding the surfactant Lipoid E80 (4.8 mg) to soybean oil (20 mL) and mixing at 60 °C under gentle stirring. The oil phase was then added dropwise to the water phase (Milli-Q water) and mixed using an immersion sonicator (Ultrasonic Processor VCX500 Sonic and Materials) for 3 min (with a sonication amplitude of 70%and a pulse on and a pulse-off duration of 10 and 5 s, respectively), followed by an additional 5 min under the same conditions. The system was thermostated with an ice chamber to prevent overheating. The pre-emulsions were then passed through the high-pressure homogenizer (Microfluidics M110PS) at 2000 bar for the first three individual cycles to greatly reduce the initial size, and the reservoir was continuously refilled for 200 steps. This method was used to prepare oil-in-water nanoemulsions at 20 wt% of the oil concentration. Next, a 0.1 M acetic acid solution of chitosan (Ct) or fluorescein isothiocyanate-labeled chitosan (Ct_FITC_) (0.125 wt%) was prepared. The O/W NE phase (2.5 mL O/W NE 20 wt% oil in 7.5 mL H_2_O) was added quickly to the chitosan solution under vigorous stirring and kept under stirring for 15 min to allow uniform chitosan deposition. The final concentrations of oil and chitosan were 1 wt% and 0.01 wt%, respectively. These nanoemulsions were redispersed using the previously reported method and stored at room temperature [10].

### 2.8. Modification of Chitosan with Fluorescein Isothiocyanate

Ct was labeled with fluorescein isothiocyanate (FITC) fluorophore based on a previously reported procedure [30]. Ct (100 mg, 0.5 mmol) was dissolved in 10 mL of a 0.1 M acetic acid solution. After complete dissolution, a solution of FITC (5.0 mg in 500 μL of DMSO) was added dropwise. The reaction proceeded overnight at room temperature, protected from light. The sample was then dialyzed (dialysis tubing of 3.5 kDa) against water multiple times over a couple of days to remove unreacted dye. Finally, the purified product was freeze-dried for 24 h. The degree of functionalization was determined with 1H NMR spectra and resulted in less than 1%.

### 2.9. Layer-by-Layer GOQDs-HA Deposition on Chitosan-Coated NE

Starting from the chitosan-coated NE (1 wt% oil and 0.01 wt% chitosan), a negatively charged GOQDs-HA or GOQDs-HA_Rhod_ second layer was deposited by mixing 1:1 (*v*/*v*) a 0.24% *w*/*v* aqueous dispersion of GOQDs-HA with the Ct-NE. The optimal GOQDs-HA concentration in the final emulsion of 0.24% *w*/*v* was determined via a saturation curve, starting from concentrations of 0.3% *w*/*v* of GOQDs down to a concentration of 0.05% *w*/*v*. The deposition was achieved through a procedure previously developed by our team, which allows for higher control over the deposition process [31]. Experimentally, the two liquid phases were injected with the aid of two syringe pumps (HARVARD APPARATUS 11 PLUS) at the same flow rate (0.4 mL min^−1^) through two micrometric capillaries interfaced at their extremities. Each drop was then collected inside a glass tube immersed in an ultrasonic bath (FALC INSTRUMENTS), at 10 °C, 40 kHz, and 100% power to obtain bilayer Ct-GOQDs-HA and Ct-GOQDs-HA_Rhod_. For control, a bilayer without GOQDs-HA was prepared using the same procedure (HA-Ct_FITC_-NEs).

### 2.10. Curcumin Loading and Release

Curcumin (Cur)-loaded GOQDs were prepared by a simple, noncovalent interaction method. Cur was dissolved in ethanol 66% *v*/*v* (4 mg/mL) and added drop by drop to the GOQDs-NH_2_ and GOQDs-HA solutions (4 mg/mL) for over 1 h while stirring in the dark. After 1 h, the products were transferred to a 2 mL Eppendorf and centrifuged at 1000 rpm for 10 min. The supernatants were kept, and the precipitates were washed 3 times with phosphate buffer saline (PBS, pH 7.4) to remove free Cur molecules. The supernatants were combined, and the amount of Cur present was calculated from a Cur calibration curve by measuring the absorbance at 425 nm. The amount of loaded Cur (%_l_) was then calculated using the following equation: ${\%}_{l}=\frac{m_{c}-m_{s}}{m_{GO}}\times 100,$ where m_c_ is the initial mass of Cur, m_s_ is the mass of Cur in the supernatants, and m_GO_ is the mass of GOQDs. The release values of Cur from the two systems were evaluated at pH 7.4 and pH 5.5 for 48 h. A release experiment was also performed at 60 °C for 5 min. Briefly, the precipitates containing Cur/GOQDs-NH_2_ and Cur/GOQDs-HA were diluted with PBS buffer (pH 7.4 and 5.5) and incubated in the dark (at 37 °C or 60 °C) under continuous moderate shaking (140 rpm). At predetermined time intervals, a portion of the sample was collected and centrifuged at 4000 rpm for 15 min. Fresh buffer (2 mL) was then added to keep the volume constant. The supernatant was dispersed in EtOH 66% *v*/*v*, and the absorbance was measured at 425 nm. The amount of Cur present in the precipitates was obtained from the UV/Vis calibration curve of Cur. The percentage of Cur released was calculated using the following equation: (R_%_$=\left(\frac{m_{2}}{m_{1}}\right)\times 100\;$, where m_1_ is the initial content of Cur and m_2_ is the amount of Cur released. All measurements were performed in triplicate, and R_%_ was plotted as a function of time.

### 2.11. Cell Culture

Experiments were performed on HDFs, MCF10a, and MDA-MB.231. HDFs were extracted from healthy breast biopsies and grown in 150 cm^2^ polystyrene tissue culture flasks in enriched minimum essential medium (MEM) Eagle supplemented with 20% fetal bovine serum (FBS), 1% L-glutamine, and 1% of penicillin/streptomycin. MDA-MB-231 cells were cultured in 75 cm^2^ polystyrene tissue culture flasks in Dulbecco’s modified Eagle’s medium (DMEM/F-12) supplemented with 10% FBS, 1% L-glutamine, and 1% penicillin–streptomycin. MCF10A cells were cultured in 75 cm^2^ polystyrene tissue culture flasks in DMEM-F12 supplemented with 5% horse serum, 1% penicillin/streptomycin, 1% L-glutamine, 20 ng/mL epidermal growth factor, 500 ng/mL human corticosteroids and insulin. The culture media were changed every 2/3 days until reaching 90% confluence. Cells were washed three times with PBS and incubated with trypsin−ethylenediaminetetraacetic acid (EDTA) (0.25% trypsin, 1 mM EDTA) for 5 min at 37 °C to detach the cells.

### 2.12. Calibration Curve for GOQDs-NH_2_, GOQDs-HA, and Cells

The standard calibration curves for all the systems (GOQDs-HA and GOQDs-HA_Rhod_, HA-Ct_FITC_-NEs, and GOQDs-HA_Rhod_-Ct-NEs) were obtained by measuring their fluorescence intensity at different known concentrations using an LS55 spectrofluorometer (Perkin Elmer). Specifically, fluorescence spectra were recorded at 370 nm excitation and 500 nm emission wavelengths for GOQDs-HA, at 349 nm excitation and 577 nm emission wavelengths for GOQDs-HA_Rhod_ and GOQDs-HA_Rhod_-Ct-NEs, and 488 nm excitation and 515 nm emission wavelengths for HA-Ct_FITC_-NEs. The standard calibration curves for HDF were obtained by culturing varying known numbers of cells in a 24-well plate. Cells were allowed to adhere for 6 h at 37 °C and then incubated with 1 mL of 1 μg/mL nuclear dye Hoechst 33,342 (Invitrogen) for 20 min. After incubation, cells were rinsed once to remove non-internalized Hoechst and incubated with 0.5 mL of ddH_2_O at 37 °C for 1 h. The samples were then transferred to a −80 °C condition for 30 min and then returned to 37 °C for 30 min. At this point, cells inflated by water were lysed with 0.5 mL of solution including 2× TNE buffer (10× TNE buffer: 100 mM Tris, 10 mM EDTA, 1.0 M NaCl, pH 7.4). Then, the samples were measured at 350 nm excitation and 500 nm emission wavelengths.

### 2.13. Quantification of Cell Internalization

To quantify nanoparticle (NP) internalization, HDF cells were cultured at a density of 40,000 cells/well in 24-well plates. 24 h after cell seeding, cells were incubated for 24 h at 37 °C with all the systems dispersed in cell culture medium at a final concentration of 0.01 mg/mL. For each NP system, cells were cultured in six different wells, three of which were used for the quantification of NP internalization and three for cell counting. After incubation, cells were rinsed five times with PBS to remove non-internalized NPs, and three wells were incubated with 1 mL of 1 μg/mL nuclear dye Hoechst 33,342 (Invitrogen) for 20 min. Then, the lysis protocol, described in the previous subparagraph, was applied so that cell lysates not stained with Hoechst dye were analyzed at the wavelengths of the corresponding NPs systems, whereas cell lysates stained with Hoechst were analyzed at 350 nm excitation and 500 nm emission wavelengths. Fluorescence intensity for NPs and Hoechst 33,342 were interpolated on the corresponding calibration curves, obtained as indicated in the previous subparagraph and converted to NP weight and cell number. Finally, the NP weight was normalized on the corresponding cell number to obtain the weight of internalized NPs per cell.

### 2.14. Co-Localization with Lysosomes

After 24 h of incubation with NP systems, HDFs were rinsed twice with PBS to remove non-internalized NPs and fixed with 4% paraformaldehyde at RT. Then, cells were permeabilized and blocked with 0.1% saponin-5% bovine serum albumin (BSA)-PBS for 1n h at RT. Mouse anti-LAMP 2 polyclonal primary antibodies and, with Alexa Fluor 488 (for GOQDs-HA_Rhod_ and GOQDs-HA_Rhod_-Ct-NEs) or Alexa Fluor 546 (for HA-Ct_FITC_-NEs), goat anti-mouse secondary antibodies (Molecular Probes, Invitrogen) were used to localize the lysosomes. All samples were finally observed using a confocal microscope (SP5 Leica) with a 63× oil immersion objective. The co-localization analysis was performed by the JACoP plugin [32] to estimate the overlap coefficient among the pixels in the dual-channel images.

### 2.15. Statistical Analysis

Data are reported as mean ± standard error (SE) unless otherwise indicated. Statistical comparisons were performed with a Student’s unpaired test. *p* values < 0.05 denote statistical significance.

### 2.16. 3D Dermis for Penetration Tests

The 3D human dermis models (HDEs) were obtained using a bottom-up tissue engineering approach, wherein 3D dermal microtissues were assembled in the maturation chamber of a perfusion bioreactor. To produce 3D dermal microtissues, human fibroblasts were cultured with gelatine porous microcarriers (GPMs) (diameter 75–150 mm) in spinner flask bioreactors [33]. Briefly, 50 mg of GPMs were loaded together with 7.5 × 10^5^ cells, corresponding to an initial ratio of 20 cells/GPM, and cultured for up to 12 days. The medium was changed on the first day and every 3 days until the end of the experiments. At the end of the spinner culture, the 3D dermal microtissues were transferred to the maturation chamber of a bioreactor and cultured for 6 weeks, as previously reported [34]. Once HDEs were withdrawn from the maturation chamber, they were distributed into two 96-well round bottom clear Ultra-Low Attachment Multiple Well Plates (Corning Costar) and placed on the Orbital Shaker (ITA^TM^) at 37 °C in a humidified 5% CO_2_ atmosphere with GOQDs-HA-based systems dispersed in cell culture medium at final concentrations of 0.1 mg/mL and 0.001 mg/mL. At three different time points: 24 h, 48 h, and 72 h, the 3D- μTP were washed with PBS and fixed with 4% paraformaldehyde (PAF 4%).

## 3. Results and Discussion

With the ultimate goal of providing a multistage nanocarrier, we started from an established stable secondary nanoemulsion: a chitosan-coated oil core, referred to as Ct-NEs. Subsequently, we developed GOQDs of uniform size conjugated with HA (Figure 1). HA was chosen in this study not only to confer biocompatibility and targeting capability against CD44, while also facilitating deposition by electrostatic interaction around the positively charged Ct-NEs during the build-up of the complete nanocarrier.

### 3.1. Amino-Modified GOQDs-HA Preparation and Characterization

Amino-functionalized GOQDs (GOQDs-NH_2_) were derived from graphene oxide (GO_Hummer_) prepared using Hummer’s method, following a procedure reported in the literature with slight modifications [29]. During the reaction, the solution color gradually shifted from dark brown to yellow, as depicted in frames illustrating the reaction progress (Appendix A). It is reported in the literature that under these conditions, GOQDs-NH_2_ formation from the GO obtained with the Hummer method occurs within 24 h at 80 °C. We monitored the progress of the reaction through fluorescence measurements, considering that reducing carbonaceous materials to the nanometric scale results in the origination of photoluminescence [35]. Indeed, we observed a gradual increase in fluorescence intensity upon excitation at 350 nm over time until it reached a maximum after 8 h, prompting us to conclude the experiment at this point (Appendix A). Then, the product was brought to room T, filtered through a 0.22 μm cellulose acetate membrane, dialyzed for 2 days against deionized water, and freeze-dried to obtain GOQDs-NH_2_ (41.8% yield). Upon dispersion in H_2_O, the GOQDs-NH_2_ formed a highly stable clear solution with no visible precipitation or aggregation phenomena (Appendix A). The obtained GOQDs-NH_2_ were then conjugated with HA by exploiting electrostatic interaction around positively charged chitosan-coated NEs to increase their biocompatibility and provide them the ability to selectively recognize cells with CD44 receptors. To form amide bonds between the carboxylic groups of HA and the amine groups of GOQDs-NH_2_, an EDC/NHS coupling technique was used. EDC activated the carboxyl groups of HA, enabling them to form amide bonds with the primary amines of GOQDs-NH_2_. Then, the carboxylic acid attacked EDC under acidic conditions, forming an O-acylisourea intermediate that is extremely reactive and short-lived in an aqueous environment. The NHS improved the stability of the activated ester and thus increased the product yield. Due to the presence of both amino and carboxyl groups in GOQDs-NH_2_, amide bonds could also form between two GOQDs-NH_2_ [36]. To reduce such GOQDs-NH_2_ cross-linking, the carboxylic activators EDC and NHS were first reacted with HA for 1 h before adding GOQDs-NH_2_. The unreacted species and unconjugated GOQDs were removed from the reaction mixture through dialysis (MWCO 12,000−14,000) against DI water (three times), and the GOQDs-HA conjugate was finally recovered by lyophilization as a brownish solid (78% yield). Morphology and size evaluation of GOQDs-HA were characterized by TEM and cryo-TEM analysis. GO_Hummer_ sheets had a quasi-hexagonal shape with dark regions due to the presence of multiple layers of graphene stacked on top of each other (Appendix A). The centrifugation step at 8000 rpm effectively allowed the separation of the smaller-size GO sheets from the bulk containing larger particles. In fact, the sample showed a size distribution between 10 and 80 nm with average lateral dimensions around 35–40 nm. After oxidative cutting, the reduction of the lateral dimensions was evident, with GOQDs-NH_2_ nanosheets assuming an almost circular shape with a lateral size of around 12 nm (Appendix A). The transparent regions indicated the exfoliation of the nanoparticles into a single layer or a few layers of graphene. After conjugation with HA, the nanosheets retained the nanometric lateral dimensions as well as the very thin thickness keeping the transparent feature but became more irregular in shape with a lateral size increase (Figure 1a), possibly due to the interconnection between GO nanosheets. Energy dispersive X-ray spectroscope (EDX) measurements were performed to analyze the relative elemental composition on the surface and to verify the degree of purity of the samples obtained. As shown in Appendix A, GO_Hummer_ was free of impurities from potassium permanganate and sulfuric acid used for oxidation. The degree of oxidation of the prepared GO_Hummer_ can be estimated from the C/O ratio. Optimally oxidized graphene oxide C/O weight ratio lies between 2.1 to 2 [37]. The samples of GO prepared in this work exhibited a C/O weight ratio of 2.15, consistent with the range reported for well-oxidized graphene. After the oxidative cutting (Appendix A), the degree of oxidation of the sample increased and in the elemental analysis, the nitrogen due to the amine functionalization appeared. Compared with GOQDs-NH_2_, the higher C and O peaks of GOQDs-HA (Figure 1d) clarify the successful conjugation with HA. The peak at 1.5 keV was due to the gold sputter coating of the samples. Atomic force microscopy (AFM) imaging confirmed the decrease in lateral dimensions. The average thickness of the GO_Hummer_ nanosheets was around 5 nm corresponding to different layers of graphene (Appendix A), whereas in the samples subjected to oxidative cutting (Appendix A) and functionalized with HA (Figure 1c), the thickness was around 1 nm corresponding to the thickness of a monolayer of graphene.

X-ray photoelectron spectroscopy (XPS) was utilized to investigate the degree of oxidation of the products obtained, as well as the composition and nature of the oxygenated functional groups present. XPS survey spectra for pristine graphite (G), GO_Hummer_, GOQDs-NH_2,_ and GOQDs-HA are depicted in Figure 2. For pristine graphite (Figure 2a), the minor O1s peak was attributed to small amounts of physio-absorbed oxygen [38]. In the case of GO_Hummer,_ an increase in oxygen concentration, coupled with a reduction in the carbon content, was observed (Figure 2b). Following oxidative cutting, the oxygen content rose to 45%, indicating effective oxidation of the sample, with the presence of N-related peaks also noted (Figure 2c). In the instance of GOQD-HA (Figure 2d), a significant nitrogen peak was observed at 402 eV, as GOQD-NH_2_ was conjugated with HA through an amide bond. Table 1 shows the atomic concentrations of C1s, O1s, and N1s, along with their ratios for the four samples.

To discern the functional groups formed on the edges and basal planes of the carbonaceous lattice after the two processes, the high-resolution C1s, O1s, and N1s core level spectra of all the samples were deconvoluted into different binding energies (B.E.), as summarized in Appendix A. The C1s spectrum of GO_Hummer_ exhibited the characteristic peak of the carbonaceous skeleton at 284.57 eV, along with the characteristic peaks of the oxygenated functional groups such as ether and epoxy (286.43 eV) and carbonyl/carboxyl (288.44 eV) groups. In the sample treated with the oxidative cutting, four types of carbon atoms were detected in the C1s high-resolution spectrum: graphitic C-C, C=C (284.55 eV), C-O functions (286.11 eV), and C=O and COOH groups (288.55 eV and 289.98 eV). However, the latter two functions may be present in GOQDs due to the weak energy difference between C-C or C=C and C-N or C=N functions. Similar peaks were observed in GOQDs-HA. Two N1s signals located at 398.99 eV (C-N and C=N) bonds and 400.50 eV (protonated NH3^+^ functions linked to carbon atoms), in the N1s spectrum of GOQDs-NH_2_ further confirmed the covalent anchorage of NH_3_ at the surface of GOQDs. The optical properties of amino-functionalized GOQDs are described in Appendix A.

### 3.2. Curcumin Loading and Release

With the large specific surface area, GOQDs are expected to exhibit excellent loading behavior. Curcumin (Cur) was chosen as a model hydrophobic active compound to analyze the loading and release capacity of pharmacologically active molecules from GOQDs-HA as well as from GOQDs-NH_2_, serving as a reference to assess the dependence of HA coating from temperature and pH. Cur was loaded onto the surface of the QDs by leveraging the hydrophobic π-π interactions between the aromatic structure of Cur and the aromatic network of the GOQDs. The Cur loading yield (%_l_) was determined from a Cur calibration curve (Appendix A) by measuring the absorbance at 425 nm. Both systems exhibited high loading efficiency, with the Cur loading ratio for GOQDs-HA slightly lower than that of GOQDs-NH_2_ (177% for GOQDs-HA and 225% for GOQDs-NH_2_), likely due to the coating with HA, which hides some aromatic groups of the QDs surface. After analyzing the loading, we evaluated the ability of GOQDs to retain Cur before being internalized in cells. Thus, we examined Cur release patterns from GQDs-HA and GOQDs-NH_2_ nanocarriers at 37 °C for 48 h at two pH values of 7.4 and 5.5, representing the physiological pH of whole-body fluid and tumor micro-environment, respectively. As depicted in Figure 3a, at pH 7.4, GOQDs-HA released only 20% of Cur in the first 10 h, with the release remaining almost constant [39]. The release of Cur from GOQDs-NH_2_ instead reaches 50% already in the first 5 h of the experiment, followed by a trend like that of GOQDs-HA. Cur release from GOQDs-HA exhibited a slower release rate compared to that from GOQDs-NH_2_, suggesting that HA could cap the GOQDs, thus slowing down Cur release. This is particularly relevant when considering the intravenous administration of nanocarriers, as surface coating could delay the early release of the active compound during blood circulation. Under acidic conditions (Figure 3b), the Cur release trend did not change significantly. Moreover, pH variation did not significantly affect Cur release, which would be beneficial in preventing an early release of the drug in the tumor microenvironment before entering into the cells.

Conversely, we assessed the ability to actively control the release of Cur by exploiting the thermal effect. As previously described, nano-sized carbon materials have a high photothermal effect under low-power near-infrared (NIR) irradiation due to their effective light-to-heat conversion compared to other carbon allotropes [40] hence they are promising candidates for photothermal therapy, a non-invasive treatment that uses photo-absorbers to generate heat from light absorption to burn cancer cells. When the light absorber also functions as a drug delivery system, it becomes possible to combine chemotherapy with phototherapy. The local temperature increase generated by irradiation in fact can elevate the release rate by breaking π-π stacking bonds at temperatures higher than 40 °C [41]. To simulate the photothermal effect, we investigated the release of Cur from GOQDs-NH_2_ and GOQDs-HA at 60 °C, over a 5 min interval (Figure 3c). Once again, GOQDs-HA showed less release at the beginning of the experiment and then the amount of Cur released increased exponentially over the next 5 min, suggesting that Cur release could be enhanced by the photothermal effect at tumor sites.

### 3.3. GOQD-HA-Coated Ct-NEs

The first step in preparing the complete nanocarrier, consisting of GO nanosheets coated with hyaluronic acid deposited on Ct-NEs (GOQDs-HA-Ct-NEs), involved preparing the nanoemulsion according to the procedure reported in the Materials and Methods section. Dynamic light scattering (DLS) analyses were conducted to verify size distribution and polydispersity index (PDI). As shown in Figure 4, DLS analysis revealed that the O/W NE had a very narrow dimensional distribution centered at 98 nm with a PDI value of 0.087. To stabilize the O/W NE, this primary nanoemulsion was coated with a layer of chitosan, a biodegradable polysaccharide [42]. To prevent aggregation, the Ct-NEs were properly re-dispersed according to the previously reported procedure [9]. The success of the Ct coating was confirmed by the switching of the ζ potential, which changed from −25.6 mV for the O/W NE to 28.1 mV for the secondary NE, Ct-NE. Ct-NE (monolayer 1%Oil_0.01%Ct) exhibited lateral dimensions similar to the primary O/W NE (Figure 4) and even higher monodispersity, as demonstrated by the lower PDI value (0.061 vs. 0.087), due to the stabilization effect of the Ct around the oil nanodroplets.

Innovatively, in this work, amino-functionalized GOQDs coated with hyaluronic acid were assembled around Ct-NEs through electrostatic interaction, allowing for precise control over the deposition process. To determine the optimal concentration of GOQDs-HA that would ensure a complete switch of the surface charge and thus a thorough coating of the Ct-NE, ensuring high stability over time, we constructed a calibration curve starting from a GOQDs-HA concentration of 0.05% (*w*/*v*) up to a concentration of 0.30% (*w*/*v*). By monitoring size (Appendix A), PDI (Appendix A), and ζ potential (Appendix A) parameters over 4320 h (=180 d), we established that the GOQDs-HA concentration of 0.12% (*w*/*v*) was sufficient to completely cover the monolayer and ensure high stability over time.

The size and ζ potential for the obtained system, Bilayer 0.5%Oil_0.05&CT_0.12%GO-HA, referred to as GOQDs-HA-Ct-NEs, are reported in Figure 5. GOQDs-HA-Ct-NEs showed high stability in aqueous solution when stored at 4 °C, as evidenced by the values of the DLS parameters recorded up to 180 days from its preparation (Figure 5 and Figure 6a), around 145 nm. The size of the bilayer was also analyzed with nanoparticle tracking analysis (NTA), one of the few methods able to visualize and measure nanoparticles in suspension in the range from 10 to 1000 nm based on the analysis of Brownian motion (Figure 6b). For that purpose, the dispersion was diluted with ultrapure water to achieve an optimal concentration range of 10^7^–10^9^ particles per ml (approximately 20–100 particles in the field of view of the NanoSight video window) [43], and it was dispersed by sonication for 10 min in an ultrasonic bath before introducing it into the NanoSight flow cell. The average size of three measurements gave a value of 112.09 ± 0.7 nm. A comparison between DLS and NTA results demonstrates that DLS may overestimate the sample particle sizes. Both approaches yield a hydrodynamic radius (Rh) value, but larger particles scatter more intensely and are better detected by DLS than smaller particles because the scattering intensity effect is proportional to particle size rather than the number of particles of equivalent diameters provided by NTA [44,45]. Because it is based on the tracking of individual particles, NTA identifies the distribution of sizes (10–2000 nm) with greater precision.

Considering the two components of the multistage nanocarrier, namely the GOQDs from one side and the Ct-NEs on the other side, co-localization with confocal microscopy analyses were performed to assess the correct assembly. For this purpose, we labeled the chitosan layer with FITC and GOQDs-HA with Rhodamine B. Rhod was conjugated to GOQDs by exploiting the electrostatic interactions of the sp^2^ carbon network with the aromatic structure of the fluorophore. Successful dye loading was verified by the fluorescence spectrum of GOQDs-HA_Rhod_ as shown in Appendix A, while the percentage of fluorophore loading was measured by UV-Vis absorption measurement at 557 nm through a Rhod calibration curve.

Multiphoton confocal analysis of the GOQD-HA_Rhod_-Ct_FITC_-NEs showed a perfect match between green and red colors proving the effectiveness of the deposition homogeneity (Figure 7).

Accordingly, the fluorescence spectra obtained by exciting at the λ of the two fluorophores confirmed the co-presence of FITC and Rhod (Appendix A).

### 3.4. Cellular Internalization and Cell Viability of GOQD-HA

To study the interaction of the multistage nanocarriers with cells, HDFs, MCF10A, and MDA-MB-231 were incubated with GOQDs-HA-Ct-NEs (0.5%Oil_0.05%CT _0.12%GOQDs) at 37 °C for 24 h. Both single GOQD-HA and the bilayer system HA-Ct-NEs were used as controls to elucidate the role of the GOQDs in the complete nanocarrier. Considering the quenching of the auto-fluorescence of the GOQDs in contact with the positive charges of Ct, for cell internalization Ct and HA in the investigated systems were labeled with fluorophores Rhod and fluorescein isothiocyanate (FITC), respectively.

Uptake experiments were performed by incubating HDF, MCF10A, and MDA-MB-231 cells with the three systems, GOQD-HA_Rhod_, HA-Ct_FITC_-NEs, and GOQDs-HARhod-Ct-NEs at a final concentration of 0.01 mg/mL. The amount of up-taken NPs was assessed by performing a spectrofluorimetric assay, as described in the Materials and Methods section (Figure 8). In HDFs, the uptake of GOQDs-HARhod-Ct-NEs was an order of magnitude higher than that of HA-Ct_FITC_-NEs and very similar to that of GOQDs-HA_Rhod_ (Figure 8b). This could be attributed to the release of the GOQDs-HA from the entire nanocarrier upon its deconstruction once in contact with cells, consistent with the multistage feature of the proposed nanocarrier. Similar results were found in MCF10A and MDA-MB-231 cells (Appendix A). Both cell lines exhibited a greater ability to internalize GOQDs-HARhod-Ct-NEs and GOQDs-HA than HA-Ct_FITC_-NEs. More interestingly, active targeting of HA decoration against CD44 overexpressed by cells [46] was evidenced by the significantly higher uptake of both GOQDs-HARhod-Ct-NEs and GOQDs-HA in breast adenocarcinoma cells (i.e., MDA-MB-231) compared to breast epithelial cells (i.e., MCF10A). It should be noted that MDA-MB-231 cells exhibited greater internalization of both GOQDs-HARhod-Ct-NEs and GOQDs-HA compared to HDF cells, as well (Appendix A). This difference becomes even more relevant considering that MDA-MB-231 cells have a smaller and comparable volume (i.e., a reduced 3D surface by which the bilayer systems can interact with cells) compared to HDFs and MCF10A, respectively (Appendix A).

Cell viability was assessed 24 h after incubation and, as reported in Figure 8i all the systems here investigated showed no significant toxic effects on the viability of HDFs although further tests are needed to study longer-term cytotoxicity.

Before testing GOQD-HA_Rhod_, cell internalization was analyzed by incubating cells with GOQD-HA with and without Rhod labeling (Appendix A) and the data clearly showed that the presence of Rhod did not alter the cellular uptake. Additionally, the absence of cytotoxicity of GOQDs-HA was previously tested in the HDF cell line (Appendix A).

To investigate the mechanisms of cellular internalization, HDF cells were incubated with the two bilayer systems HA-Ct_FITC_-NEs and GOQDs-HARhod-Ct-NEs. After 24 h of incubation, lysosomes were localized by treatment with Mouse anti-LAMP 2 polyclonal primary antibodies and, with Alexa Fluor 488 goat anti-mouse secondary antibodies. HDF cells were observed using a confocal microscope. Representative confocal images of HDFs incubated with the two systems are reported in Figure 9. The colocalization analysis between LAMP2 and HA-CT_FITC_-NEs/GOQDs-HA_Rhod_-Ct-NEs was performed in terms of Meander’s overlap coefficient M1 [47]. This coefficient, which can assume values from 0 to 1, provides information on the percentage of bilayer systems entrapped in lysosomes. It was found to be almost 0.6 for HA-CT_FITC_-NEs and ~0.2 for GOQDs-HA_Rhod_-Ct-NEs, indicating that most of HA-CT_FITC_-NEs were localized in the lysosomes, whereas most GOQDs-HA_Rhod_-Ct-NEs were free in the cytoplasm or trapped in structures other than lysosomes. Further investigations are needed to better elucidate the uptake mechanism of these systems or their fate after having entered the cell membrane.

### 3.5. Penetration Analyses in HDE

HDE are biohybrid tissues obtained through a bottom-up tissue engineering approach that involves dynamic cell seeding of human dermal fibroblasts on porous gelatine microcarriers using a spinner flask bioreactor [48]. HDEs represent a highly physiologically relevant in vitro model of their native counterpart. They are formed by fibroblasts embedded in their own ECM, presenting both collagenous and non-collagenous molecules correctly assembled and organized as in the in vivo dermis [34]. Therefore, they are ideal models for studying tissue penetration of engineered drug delivery systems. To localize GOQDs-HA within the tissue, we labeled GOQDs-HA with Rhod. As in the case of Cur, Rhod was conjugated to GOQDs by exploiting the electrostatic interactions of the sp^2^ carbon network with the aromatic structure of the fluorophore. Successful dye loading was verified by the fluorescence spectrum of GOQDs-HA_Rhod_, as shown in Appendix A, while the percentage fluorophore loading was measured by UV-Vis absorption at 557 nm, using a Rhod calibration curve (Appendix A). To analyze the spreading of the GOQDs-HA in the tissue, they were put in contact with GOQDs-HA_Rhod_ and GOQDs-HA_Rhod_-Ct-NEs, at a concentration of 0.1 mg/mL. The tissues were then washed, and multiphoton confocal microscopy was used to assess HDE’s collagen network via SHG imaging and nanocarrier penetration into the tissue by exploiting Rhodamine fluorescence. After careful removal of the background, the collected fluorescence intensities indicated that the fluorescence intensity decreases with increasing distance in both systems both at time 0 and two hours after the contact with HDE. The fluorescence intensity of both systems doubled after 2 h of contact. Interestingly, GOQDs-HA_Rhod_-Ct-NEs at time 0 showed lower penetration than the GOQDs-HA_Rhod_, while over time, at *t* = 2 h, it showed a penetration behavior similar to that of GOQDs-HA_Rhod_ (Figure 10). This is expected in the case of degradation of the overall nanocarrier followed by the release of the smaller nanoparticles based on GOQDs-HA, according to the multistage approach we are pursuing.

## 4. Conclusions

Nanoparticles have the potential to revolutionize modern therapies for many diseases, including cancer. However, when rationally designing a system for nanomedicine, both the heterogeneous and complex tumor microenvironment and the physiological barriers of the organism must be carefully considered. Generally, systems with dimensions greater than 100 nm can selectively accumulate in tumor tissues by exploiting the high permeability of a growing tumor mass but have poor diffusion in the dense collagen matrix of the interstitial space, resulting in restrictive nanoparticle accumulation around tumor blood vessels and little penetration into the tumor parenchyma. On the other hand, 10–30 nm nanoparticles show high cellular uptake, as well as higher diffusion in the extracellular matrix, but can be easily sequestered by the mononuclear phagocytic system and eliminated. To overcome this problem, we have designed a multistage nanocarrier, where GOQDs-HA were deposited on an O/W emulsion coated with a layer of chitosan, Ct-NEs, through a layer-by-layer technique optimized for liquid templates. The GOQDs-HA concentration required to stabilize the bilayer was determined through a saturation curve starting from concentrations of 0.05% *w*/*v* of QDs up to a concentration of 0.30% *w*/*v*. The bilayer obtained with a GOQDs-HA concentration of 0.12% was selected for the best stability shown up to 180 days, as evidenced by the measurements of the size at the nano tracking and the ζ potential at DLS. In vitro tests showed no evident cytotoxicity toward HDF cell lines despite the huge uptake shown in both cell lines. Moreover, GOQDs-HA showed similar uptake whether it was free or coated on Ct-NEs may be due to the destructuring of the complete GOQDs-HA-Ct-NEs nanocarrier once in contact with the cell. Additionally, preliminary penetration experiments in three-dimensional microtissues confirmed the ability of the GOQDs-HA to enter within the tissue even when coated on Ct-NEs, in agreement with the GOQDs-HA-Ct-NEs destructuring, according to the proposed multistage approach.

## Data Availability

The original contributions presented in the study are included in the article/supplementary material, further in-quiries can be directed to the corresponding author/s.

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
