# Peer review of "Multistage Nanocarrier Based on an Oil Core–Graphene Oxide Shell"

_pharmaceutics, 2024, doi:10.3390/pharmaceutics16060827_

Round 1

Reviewer 1 Report

Comments and Suggestions for Authors

1. Throughout the manuscript: For ‘Fig.X’, leave a space between ‘Fig.’ and ‘X’.

2. L421: Change ‘Fig 2C’ to ‘Fig. 2C’

3. In Fig. 2 and Fig. 3: a), b), c), and d) are missing.

4. In Fig. 3: In the release results of the left-side at upper in Figure 3, can the percentage of Cur released (R%) exceed 100% for the GOQDs sample? I think that it is impossible.

5. In Fig. 3: In the release results of the right-side at upper in Figure 3, Why does the release amount of the drug suddenly increase after 70 h?

6. In the caption of Fig. 3: The caption is not clear. Please clearly describe the release condition. Also, describe the content related to Fig. 3 in the text with the clear release condition.

7. Why is the drug released quickly at 60 oC? Please describe in detail in the text.

8. In Figure 4: Insert the title and unit of the X-axis and Y-axis.

9. In Figure 6: Make the font size larger in Figures 6a and 6b. And delete the general measurement information shown at the top of the graph in Figure 6a.

10. Throughout the manuscript: Please check again for typos and awkward phrases.

-----------------------------------------------The end----------------------------------------

Comments on the Quality of English Language

This manuscript was generally well-written in English. But, please check again for typos and awkward phrases.

Author Response

  1. Throughout the manuscript: For ‘Fig.X’, leave a space between ‘Fig.’ and ‘X’. Done
  2. L421: Change ‘Fig 2C’ to ‘Fig. 2C’ Done
  3. In Fig. 2 and Fig. 3: a), b), c), and d) are missing. Done
  4. In Fig. 3: In the release results of the left-side at upper in Figure 3, can the percentage of Cur released (R%) exceed 100% for the GOQDs sample? I think that it is impossible.
  5. In Fig. 3: In the release results of the right-side at upper in Figure 3, Why does the release amount of the drug suddenly increase after 70 h?

We thank the Reviewer for his/her comments (4 and 5). We acknowledge that above 48 hours there are unexpected results including R% >100 and sudden increase of the curcumin release. Considering we did not have sufficient statistics of the data above 48 hours we excluded these data in the revised version of the manuscript. In fact, our focus was on identifying release trends, verifying pH dependence, HA coating role, as well as temperature influence and it was enough to make such evaluations with the data up to 48 hours. In this way, we demonstrated that hyaluronic acid coating could cap the GOQDs slowing down Cur release and that acidic pH does not induce an unwished early release.

  1. In the caption of Fig. 3: The caption is not clear. Please clearly describe the release condition. Also, describe the content related to Fig. 3 in the text with the clear release condition. Done
  2. Why is the drug released quickly at 60 oC? Please describe in detail in the text.

We explained the release at high temperatures in the text and inserted details in the caption of Figure 3.

  1. In Figure 4: Insert the title and unit of the X-axis and Y-axis. Done
  2. In Figure 6: Make the font size larger in Figures 6a and 6b. And delete the general measurement information shown at the top of the graph in Figure 6a. Done
  3. Throughout the manuscript: Please check again for typos and awkward phrases. Done

Reviewer 2 Report

Comments and Suggestions for Authors

The Results and Discussion section is well described. In the Methods section, a scheme with all the synthesis steps should be presented. 

Author Response

The Results and Discussion section is well described. In the Methods section, a scheme with all the synthesis steps should be presented. 

We thank the Reviewer for his/her comment, we inserted a schematic of the nanocarrier synthesis steps in the result and discussion session.

Reviewer 3 Report

Comments and Suggestions for Authors

This work presents a multistage nanocarrier based on oil core-graphene oxide shell functionalized with hyaluronic acid (HA) for cancer targeting and loaded with curcumin as a drug. The authors clearly demonstrated synthesis and characterization of graphene oxide nanosheets based quantum dots (GOQDs) functionalized with HA and deposited in oil in water nanoemulsion (O/W NE). They also attempted to assess the cytotoxicity, cell internalization, lysosome co-localization and dermal penetration of the nanocarrier. The rationale behind the conceptual design of the nanocarrier is confusing and while the experimental scope is motivated, it is not supportive of intended anti-cancer application. Below are some major comments for improving the work:  

11. The motivation behind the NP design is unclear. In abstract (Lines 17-18), authors state that the NP size of “100 nm is too large for an effective penetration”, but still proceed with the synthesis of emulsion-based NPs larger than 100 nm. The idea is rather self-conflicting, as larger NPs are cleared by liver and spleen that also precludes tumor penetration.  

22. The abstract, the authors suggest that their nanocarrier is designed for tumors (Lines 22-26: “GOQDs-HA as a biodegradable core nanocarrier to degrade once in tumor, curcumin as anticancer drug”), but they fail to show data in cancer cells in vitro or in vivo; instead, they focus on human dermal fibroblasts (HDF). In the introduction (Line 85), the authors mention that “biological assessment was also performed on mammary adenocarcinoma cells, MCF-7", which is not provided.

33. The rationale behind the use of HDF and 3D dermis is not consistent with the conceptual design of this paper. In the abstract (Lines 14-16), the authors discuss blood circulation and leaky regions of tumours as well as in Section 3.2 (Lines 464-465) they consider the possibility of i.v. injection of nanocarriers, but again the focus on human dermal fibroblasts and 3D dermis penetration is confusing. If an i.v. injection is the administration route of NPs, the stability, release and degradation tests in serum should be performed. Will the NPs be delivered through skin?

44. In the abstract, the authors claim that “choice of the modification with HA is to promote an active targeting to cancer cells” (Lines 25-26), yet the cancer cell targeting ability of the NP is not supported or shown by their data.  

55. The authors claim that the advantage of using this NP is in the deconstruction of the multistage nanocarrier once in the tumor. However, there is no direct evidence showing this decomposition. They suggest that GOQDs-HA showed similar uptake independent of Ct-NE coating “maybe due to the destructuring of the complete GOQDs-HA-Ct-NEs nanocarrier once in contact with cell”, but when and whether this decomposition actually happens is not supported by experimental data (Lines 650-651). Can authors perform the cellular experiments with fluorescently labelled GOQD-HARhod and CtFITC-NEs and show that this decomposition takes place in tumour cells?

66. In Fig. 1b, the AFM image of nanocarrier morphology is not clear. Fig. S2a and S2b are not clear. Can the contrast or resolution be improved? In Fig. 7, can the assembly of GOQD-HARhod-CtFITC-NEs be visualised by TEM imaging?

77. The conditions for curcumin release study are not representative of the complex tumour environment. The authors claim that at pH 7.4, GOQDs-HA undergo continuous constant release followed by drastic increase after 70 hours, which is not consistent with data, presented in Fig. 3a. Moreover, the stability of carriers should be tested in the relevant conditions (blood, serum, pH) as well as their degradation rate.

88. Cellular internalization data is not reliable as the authors did not present images of NPs in the cells. The authors should also explore the toxicity of NPs at longer time points (Fig. 8).

99. For lysosome co-localization, bio-TEM should also be considered to demonstrate that NPs are free in the cytosol or entrapped in other organelles. Also, from Fig. 9, it seems that the NP and lysosomes colocalize. A clear overlay image should be provided. More time points should be included in this experiment, and more cell images should be provided in Supplementary Information.

110.  In general, the scope of biological assessments is limited to draw conclusions of the potential of this nanocarrier for use in cancer treatment.

Comments on the Quality of English Language

The English quality is adequate. 

Author Response

This work presents a multistage nanocarrier based on oil core-graphene oxide shell functionalized with hyaluronic acid (HA) for cancer targeting and loaded with curcumin as a drug. The authors clearly demonstrated synthesis and characterization of graphene oxide nanosheets based quantum dots (GOQDs) functionalized with HA and deposited in oil in water nanoemulsion (O/W NE). They also attempted to assess the cytotoxicity, cell internalization, lysosome co-localization and dermal penetration of the nanocarrier. The rationale behind the conceptual design of the nanocarrier is confusing and while the experimental scope is motivated, it is not supportive of intended anti-cancer application. Below are some major comments for improving the work:  

  1. The motivation behind the NP design is unclear. In abstract (Lines 17-18), authors state that the NP size of “100 nm is too large for an effective penetration”, but still proceed with the synthesis of emulsion-based NPs larger than 100 nm. The idea is rather self-conflicting, as larger NPs are cleared by liver and spleen that also precludes tumor penetration.  

We thank the Reviewer for his/her consideration. We have now clarified this concept when we introduce our nanocarrier referring to the literature cited in ref.3 which explains that 100 nm is an ideal size for extravasation by exploiting the well-known EPR mechanism but then it is needed a much smaller size for deep penetration in the tumour to reach and treat all the cells.

  1. The abstract, the authors suggest that their nanocarrier is designed for tumors (Lines 22-26: “GOQDs-HA as a biodegradable core nanocarrier to degrade once in tumor, curcumin as anticancer drug”), but they fail to show data in cancer cells in vitro or in vivo; instead, they focus on human dermal fibroblasts (HDF). In the introduction (Line 85), the authors mention that “biological assessment was also performed on mammary adenocarcinoma cells, MCF-7", which is not provided.

We thank the Reviewer for his/her constructive comment that we consider important and well-grounded. As we hope to have better explained in the Introduction, we performed our analysis in human dermis fibroblasts and in 3D biohybrid systems (i.e., HDE), because in many solid tumors, such as breast adenocarcinoma, therapeutics show a limited efficacy because of their limited penetration into the tumor stroma, composed of an abundant extracellular matrix and various stroma cells, among which fibroblasts. Therefore, we considered important to assess the ability of GOQDs-HA coated Ct-NEs to interact with stroma cells (HDFs) and to penetrate the tumor stroma surrounding tumor cells, that in our work is mimicked by HDE.

However, we retain that the need to validate the efficacy of nanocarriers, which have been designed for tumor cells, is crucial. To this end, we performed, in agreement with the Reviewer comment, new experiments in two additional cell lines, MCF10A and MDA-MB-231, that are mammary epithelial and adenocarcinoma cells, respectively. The results are reported in the revised version of the manuscript.

  1. The rationale behind the use of HDF and 3D dermis is not consistent with the conceptual design of this paper. In the abstract (Lines 14-16), the authors discuss blood circulation and leaky regions of tumours as well as in Section 3.2 (Lines 464-465) they consider the possibility of i.v. injection of nanocarriers, but again the focus on human dermal fibroblasts and 3D dermis penetration is confusing. If an i.v. injection is the administration route of NPs, the stability, release and degradation tests in serum should be performed. Will the NPs be delivered through skin?
  2. In the abstract, the authors claim that “choice of the modification with HA is to promote an active targeting to cancer cells” (Lines 25-26), yet the cancer cell targeting ability of the NP is not supported or shown by their data.  

As reported in the response to the comment 22, we tried to hit the goal, that is the active targeting to cancer cells, by assessing the ability of the nanocarriers to penetrate mammary adenocarcinoma cells (i.e., MDA-MB-231).  We found that MDA-MB-231 cells can internalize all the three systems, but more importantly their internalization ability is significantly more pronounced than that of their healthy counterparts (i.e., MCF10A) and of HDFs.

  1. The authors claim that the advantage of using this NP is in the deconstruction of the multistage nanocarrier once in the tumor. However, there is no direct evidence showing this decomposition. They suggest that GOQDs-HA showed similar uptake independent of Ct-NE coating “maybe due to the destructuring of the complete GOQDs-HA-Ct-NEs nanocarrier once in contact with cell”, but when and whether this decomposition actually happens is not supported by experimental data (Lines 650-651). Can authors perform the cellular experiments with fluorescently labelled GOQD-HARhodand CtFITC-NEs and show that this decomposition takes place in tumour cells?

We thank the Reviewer for his/her comment. It should be considered that the multistage strategy behind the design of the proposed nanocarrier is meant to solve the issue of low penetration in the tumor tissue and therefore the difficulty to treat all the cancer cells especially the ones more distant from the blood vessels. We expect that most of the cells will be directly in contact with the smaller component of the carrier, namely GOQD-HARhod and only occasionally with the entire nanocarrier. Thus, we tried to make a deconstruction experiment in the tissue but the fluorescence of FITC was overshadowed by tissue autofluorescence, and only the signal from Rhodamine was visible. Bio-TEM analyses of the tissue has been planned to assess this aspect. We will also take in consideration the suggestion for further studies on the cell internalization mechanism of the complete carrier.

  1. In Fig. 1b, the AFM image of nanocarrier morphology is not clear. Fig. S2a and S2b are not clear. Can the contrast or resolution be improved? In Fig. 7, can the assembly of GOQD-HARhod-CtFITC-NEs be visualised by TEM imaging?

We thank the Reviewer for his/her comment. The AFM images shown are at maximum resolution and contrast. We tried to image the complete carrier by TEM but was not successful, it will require a dedicated optimization work to avoid aggregation and select optimal contrasting agents to improve contrast.

  1. The conditions for curcumin release study are not representative of the complex tumour environment. The authors claim that at pH 7.4, GOQDs-HA undergo continuous constant release followed by drastic increase after 70 hours, which is not consistent with data, presented in Fig. 3a. Moreover, the stability of carriers should be tested in the relevant conditions (blood, serum, pH) as well as their degradation rate.

We thank the Reviewer for the relevant comment. We have now explained in the text that the presented curcumin release tests in the medium are of course not representative of the complex tumour environment although they were able to elucidate some dependences from the HA coating, from T and pH. Regarding the second comment the text has been updated. Finally, for what concerns the stability of the carrier at different conditions we agree that, despite most of the stabilities in literature are carried out in PBS, one should go deeper in the stability analysis in blood before moving to in vivo tests that we have planned for the next work. This time we just monitored the influence of PBS and of the pH on the stability.

  1. Cellular internalization data is not reliable as the authors did not present images of NPs in the cells. The authors should also explore the toxicity of NPs at longer time points (Fig. 8).

We thank the Reviewer for his/her comment. To make internalization data more reliable, we reported some representative images of all three systems in HDFs (Figure 8 a-c). We agree that to have a complete indication of nanocarrier safety some additional tests are needed, and we have already planned both longer cytotoxicity tests as well as some genotoxicity tests. In agreement with the Reviewer comment we have added this plan in the Results section.

  1. For lysosome co-localization, bio-TEM should also be considered to demonstrate that NPs are free in the cytosol or entrapped in other organelles. Also, from Fig. 9, it seems that the NP and lysosomes colocalize. A clear overlay image should be provided. More time points should be included in this experiment, and more cell images should be provided in Supplementary Information.

We thank the Reviewer for his/her valid comment. Nevertheless, in this work we are mainly interested to demonstrate the ability of NPs to be up taken by the cells with no relevant toxic effects. To this regard, the results of colocalization analysis demonstrate that NPs are in the cells, partially entrapped into the lysosomes. However, we aim to better describe the mechanisms of internalization in a future work, including more time points in both 2D and 3D biological systems. Furthermore, we added a merge image of lysosomes and NPs in the new version of Figure 9.

  1. In general, the scope of biological assessments is limited to draw conclusions of the potential of this nanocarrier for use in cancer treatment.

Reviewer 4 Report

Comments and Suggestions for Authors

Dear Editor, 

I have completed the review of the manuscript titled "Multistage Nanocarrier Based on Oil Core— Graphene Oxide Shell," which was submitted for publication in Pharmaceutics. I thank to the opportunity to contribute to the peer-review process. Overall, the manuscript presents an interesting study on the development of a nanoemulsion-bassed nanocarrier system, but several revisions are necessary to enhance clarity and improve the quality of the presentation. Below are my comments and suggestions for improvement:

-The peaks other than carbon and oxygen in the EDX spectra should be lightened or clarified to improve readability and interpretation.

- Figures 3 and S6 are not clear and need to be improved for better visualization and understanding of the experimental results.

- Please ensure that the axis names in Figure 4 are clearly labeled for easy interpretation of the data.

-It is essential to state the equation and expressions for R^2 of the calibration chart used in the study to provide transparency and reproducibility of the experimental results.

- Pay attention to the use of abbreviations and grammar throughout the manuscript to ensure consistency and clarity in the presentation of the research findings.

- Discuss the stability of the water-oil nanoemulsion and consider the necessity of a stabilizing chemical agent to maintain its stability. Additionally, provide clear experimental parameters for the sonication step, including time, temperature, and frequency.

- Explain how the amount of unencapsulated Cur was determined in the study to ensure transparency and reproducibility of the experimental procedures.

- Provide an explanation of the loading and release of Cur using FTIR analysis, focusing on relevant functional groups to elucidate the mechanisms involved in drug encapsulation and release.

- The authors cite relevant studies below to strengthen the discussion and contextualize their findings.

https://doi.org/10.1016/j.colsurfa.2022.128349

https://doi.org/10.1016/j.carbpol.2021.118174

https://doi.org/10.1007/s00289-021-03839-y

https://doi.org/10.1016/j.ijbiomac.2021.05.044

Overall, addressing these comments and revisions will significantly improve the clarity, accuracy, and comprehensibility of the manuscript. I recommend that the authors carefully address each point before resubmitting the revised manuscript for further consideration.

Best regards

Comments on the Quality of English Language

Dear Editor, 

I have completed the review of the manuscript titled "Multistage Nanocarrier Based on Oil Core— Graphene Oxide Shell," which was submitted for publication in Pharmaceutics. I thank to the opportunity to contribute to the peer-review process. Overall, the manuscript presents an interesting study on the development of a nanoemulsion-bassed nanocarrier system, but several revisions are necessary to enhance clarity and improve the quality of the presentation. Below are my comments and suggestions for improvement:

-The peaks other than carbon and oxygen in the EDX spectra should be lightened or clarified to improve readability and interpretation.

- Figures 3 and S6 are not clear and need to be improved for better visualization and understanding of the experimental results.

- Please ensure that the axis names in Figure 4 are clearly labeled for easy interpretation of the data.

-It is essential to state the equation and expressions for R^2 of the calibration chart used in the study to provide transparency and reproducibility of the experimental results.

- Pay attention to the use of abbreviations and grammar throughout the manuscript to ensure consistency and clarity in the presentation of the research findings.

- Discuss the stability of the water-oil nanoemulsion and consider the necessity of a stabilizing chemical agent to maintain its stability. Additionally, provide clear experimental parameters for the sonication step, including time, temperature, and frequency.

- Explain how the amount of unencapsulated Cur was determined in the study to ensure transparency and reproducibility of the experimental procedures.

- Provide an explanation of the loading and release of Cur using FTIR analysis, focusing on relevant functional groups to elucidate the mechanisms involved in drug encapsulation and release.

- The authors cite relevant studies below to strengthen the discussion and contextualize their findings.

https://doi.org/10.1016/j.colsurfa.2022.128349

https://doi.org/10.1016/j.carbpol.2021.118174

https://doi.org/10.1007/s00289-021-03839-y

https://doi.org/10.1016/j.ijbiomac.2021.05.044

Overall, addressing these comments and revisions will significantly improve the clarity, accuracy, and comprehensibility of the manuscript. I recommend that the authors carefully address each point before resubmitting the revised manuscript for further consideration.

Best regards

Author Response

-The peaks other than carbon and oxygen in the EDX spectra should be lightened or clarified to improve readability and interpretation.

We thank the Reviewer for his/her valid comment. The peaks other than Carbon, Oxygen and Nitrogen were due to the materials of the stub and to the metal coating for the preparation of the sample and they are Aluminium (1.5 keV), Gold (2.1 keV) and Copper (0.9 keV).

- Figures 3 and S6 are not clear and need to be improved for better visualization and understanding of the experimental results. Done

- Please ensure that the axis names in Figure 4 are clearly labeled for easy interpretation of the data. Done

-It is essential to state the equation and expressions for R^2 of the calibration chart used in the study to provide transparency and reproducibility of the experimental results. Done

- Pay attention to the use of abbreviations and grammar throughout the manuscript to ensure consistency and clarity in the presentation of the research findings. Done

- Discuss the stability of the water-oil nanoemulsion and consider the necessity of a stabilizing chemical agent to maintain its stability. Additionally, provide clear experimental parameters for the sonication step, including time, temperature, and frequency.

The peculiar feature of our basic oil in water nano-emulsion is its unprecedented stability even without stabilizing chemical agents, published for the first time in 2014 (R. Vecchione Nanoscale). Since then, such nanoemulsion has been coated with several materials including polymers and inorganic materials such as silica. This time it has been demonstrated the ability to build a multistage nanocarrier where the external shell is made of GOQDs-HA. At least for all the monitoring times it proved to be stable. Regarding the process parameters, the oil and the aqueous phases were mixed with an immersion sonicator during the first 3 min (a sonication amplitude of 70%; a pulse on and a pulse-off respectively of 10 and 5 s) and then for another 5 min under the same conditions. The system was thermostated with an ice chamber to prevent overheating.

- Explain how the amount of unencapsulated Cur was determined in the study to ensure transparency and reproducibility of the experimental procedures.

We thank the Reviewer for his/her comment. We added the calibration curve in Supporting Information.

- Provide an explanation of the loading and release of Cur using FTIR analysis, focusing on relevant functional groups to elucidate the mechanisms involved in drug encapsulation and release.

We thank the Reviewer for his/her valuable suggestion. We agree that implementing FTIR analysis in our future studies will significantly enhance our understanding of the molecular interactions governing drug loading and release. We are committed to incorporating this technique including the appropriate cells for FTIR analysis in the liquid phase to provide a more comprehensive analysis of our nanocarrier system.

- The authors cite relevant studies below to strengthen the discussion and contextualize their findings.

https://doi.org/10.1016/j.colsurfa.2022.128349

https://doi.org/10.1016/j.carbpol.2021.118174

https://doi.org/10.1007/s00289-021-03839-y

https://doi.org/10.1016/j.ijbiomac.2021.05.044

We thank the Reviewer for his/her suggestion. We have included new references in the revised version of the manuscript. 

Overall, addressing these comments and revisions will significantly improve the clarity, accuracy, and comprehensibility of the manuscript. I recommend that the authors carefully address each point before resubmitting the revised manuscript for further consideration.

Round 2

Reviewer 1 Report

Comments and Suggestions for Authors

The authors' responses to the reviewer's questions were thought to be clear and conscientious. And it seems to have been edited to reflect those contents well in the text. Therefore, this manuscript is considered acceptable in this journal

Reviewer 3 Report

Comments and Suggestions for Authors

In this revision, the authors have adequately addressed my previous concerns. 

Comments on the Quality of English Language

The English quality is acceptable. 

Reviewer 4 Report

Comments and Suggestions for Authors

Dear Editor,

 I am writing to formally recommend the acceptance of the revised manuscript titled " 
Multistage nanocarrier based on oil core - graphene oxide shell " following its revision by the authors in response to the initial review process. The revisions made have significantly improved the quality and clarity of the manuscript, and I believe the study now makes a valuable contribution to the field. Thank you for considering my recommendation. I trust that the publication of this manuscript will enrich the journal and contribute to advancing the scientific discourse in the relevant field.